# Bulk valley transport and Berry curvature spreading at the edge of flat bands

Subhajit Sinha [1,4], Pratap Chandra Adak [1,4✉], R. S. Surya Kanthi[1], Bheema Lingam Chittari[2], L. D. Varma Sangani[1], Kenji Watanabe [3], Takashi Taniguchi [3], Jeil Jung[2] & Mandar M. Deshmukh [1✉]

2D materials based superlattices have emerged as a promising platform to modulate band structure and its symmetries. In particular, moiré periodicity in twisted graphene systems produces flat Chern bands. The recent observation of anomalous Hall effect (AHE) and orbital magnetism in twisted bilayer graphene has been associated with spontaneous symmetry breaking of such Chern bands. However, the valley Hall state as a precursor of AHE state, when time-reversal symmetry is still protected, has not been observed. Our work probes this precursor state using the valley Hall effect. We show that broken inversion symmetry in twisted double bilayer graphene (TDBG) facilitates the generation of bulk valley current by reporting experimental evidence of nonlocal transport in a nearly flat band system. Despite the spread of Berry curvature hotspots and reduced quasiparticle velocities of the carriers in these flat bands, we observe large nonlocal voltage several micrometers away from the charge current path — this persists when the Fermi energy lies inside a gap with large Berry curvature. The high sensitivity of the nonlocal voltage to gate tunable carrier density and gap modulating perpendicular electric field makes TDBG an attractive platform for valley-twistronics based on flat bands.

[1] Department of Condensed Matter Physics and Materials Science, Tata Institute of Fundamental Research, Mumbai 400005, India. [2] Department of Physics, University of Seoul, Seoul 02504, Korea. [3] National Institute for Materials Science, 1-1 Namiki, Tsukuba 305-0044, Japan. [4]These authors contributed equally: Subhajit Sinha, Pratap Chandra Adak. ✉email: pratapchandraadak@gmail.com; deshmukh@tifr.res.in

The advancement in twistronics has opened up new avenues to study electron correlations physics, such as Mott insulator states[1–3], superconductivity[3,4], and orbital ferromagnetism[5,6] in twisted bilayer graphene (TBG). Recent experiments in twisted double bilayer graphene (TDBG)[7–11] and trilayer graphene aligned to hexagonal boron nitride (hBN)[12–14] also reveal correlation effects. While low-energy flat bands enhance electronic correlations[15–17], such moiré systems support topological bands with nonzero valley Chern number[5,6,14,18,19]. In fact, the observation of anomalous Hall state in TBG[5,6] has been explained by spontaneous symmetry breaking of degenerate Chern bands[20,21]. Such observations point to rich topology in twisted systems governed by nonzero Berry curvature and understanding these topological aspects is currently at the frontier[22–24]. We present direct experimental evidence that the degenerate Chern bands at K and K′ support bulk valley transport due to Berry curvature hotspots in these flat bands, an aspect that has been little explored. Valleytronics-based devices have shown immense potential[25] and twistronics-based valleytronics devices could have additional functionality.

When inversion symmetry is broken, two-dimensional honeycomb lattices with time-reversal symmetry can have nonzero Berry curvature of same magnitude, but opposite sign in two degenerate valleys, K and K′. The nonzero Berry curvature can manifest itself in bulk valley transport via valley Hall effect (VHE), as electrons from two valleys are deflected to two opposite directions perpendicular to the in-plane electric field[26,27]. In systems such as graphene with small intervalley scattering, the valley current can be detected by an inverse VHE at probes away from the charge current path in the form of a nonlocal resistance[28–30]. Pure bulk valley current has been generated and detected in moiré system of monolayer graphene aligned to hBN[28]. Similar nonlocal response has been observed in insulating systems like gapped bilayer graphene[29,30] with the insensitivity to device edge details, suggesting bulk transport. In both the systems, nonlocal resistance has been observed near the Berry curvature hotspots. In a recent study, the nonlocal resistance reported in TBG has been attributed to high-dimensional topology, as the symmetry enforces Berry curvature to be zero[31]. However, the Berry curvature mediated bulk valley transport remains to be explored in a flat band system.

In this work, we investigate TDBG where two copies of Bernal-stacked bilayer graphene are put on top of each other with a small twist angle. TDBG is a unique flat band system having electric field tuned isolated valley Chern bands[19,32–36]. While TDBG inherits electric field tunability form bilayer graphene, the moiré periodicity opens up secondary gaps, thus isolating the moiré bands and decoupling the two K and K′ valleys[18]. Together with the flatness and the electric field tunability, the finite Berry curvature associated with the Chern bands makes this system an interesting platform for hosting the valley current. In particular, TDBG enables realizing two functionalities in a single system—hosting valley current by isolated topological bands, and the control over valley current by the electric field. We measure multiple TDBG devices and observe large nonlocal resistance whenever the Fermi energy lies in the gap—the charge neutrality point (CNP) gap or the moiré gaps. We explore the dependence of the nonlocal resistance on electric field, charge density, and temperature in detailed measurements. Our analysis finds evidence that the nonlocal resistance originates from bulk valley transport, while at low temperature edge transport starts playing a role. Twistronic system, like the one we present, offers two key knobs for bulk valleytronics—firstly, the magnitude of Berry curvature is inversely related to the gap, and secondly, the tunability of Fermi velocity tunes the sharpness of the Berry curvature hotspot.

## Results

**Nonlocal transport measurement scheme and the band structure.** For detecting bulk valley current, we follow a measurement scheme similar to that used for detecting spin current in spintronics devices[37,38], as shown in Fig. 1a. A finite charge current $I$ is passed using two local probes at two opposite sides of the device channel. VHE drives a valley current along the channel and a voltage, $V_{NL}$ is generated in the nonlocal probes by inverse VHE. We quantify this as nonlocal resistance $R_{NL} = V_{NL}/I$. We independently control both the charge density $n$ and the perpendicular electric displacement field $D$ aided by the dual-gated structure of our devices, using a metal top gate and highly doped silicon back gate (see "Methods").

The perpendicular electric field has a profound effect on the band structure of TDBG[7–11]. As depicted in Fig. 1b, at zero electric field, the system has low-energy flat bands separated from higher energy dispersing bands by two moiré gaps. As the electric field is increased, a gap opens up at the CNP separating two flat bands. The moiré gaps close sequentially upon further increase of the electric field. In Fig. 1c, we present a schematic of the band structure at finite electric field and show the existence of Berry curvature hotspots in the flat bands. The color scale plot of calculated Berry curvature in Fig. 1d depicts the locations of hotspots in the **k**-space of the conduction band for K valley. Details of band structure and Berry curvature map is provided in Supplementary Note 1 (see Supplementary Figs. 1–3).

**Local and nonlocal transport at low temperature.** We now present the experimental results for a TDBG device with twist angle 1.18°. This device shows a high degree of twist angle homogeneity $\delta\theta \sim 0.05°$ over eight microns; this is crucial for observing nonlocal resistance (see Supplementary Fig. 5 and Supplementary Note 3). In Fig. 2a, we show a color scale plot of four-probe local resistance as a function of perpendicular electric field and charge density. We see large resistance at $n = 0$ at high electric field due to gap opening at CNP, and at $n = \pm n_S = \pm 3.2 \times 10^{12} \, \text{cm}^{-2}$ corresponding to the moiré gaps. Here $n_S$ is the number of electrons required to fill one flat band. In Fig. 2b, we plot the measured nonlocal resistance which is large only at the gaps. Apart from the resistance peak at the gaps, there are other high-resistance regions in the local resistance, characteristics to small-angle TDBG[7–11]. Such examples are the cross-like feature originating at $D = 0$ in the hole side and the ring-like regions in the electron side for $|D|/\epsilon_0 \sim 0.3 \, \text{V nm}^{-1}$. The absence of these features in the nonlocal signal provides evidence that the nonlocal signal is distinct from the local resistance and is only appreciable when the Fermi energy crosses the gaps that possess large Berry curvature.

In Fig. 2c–e, we plot line slices from the color plots to show both the local and nonlocal resistances as a function of charge density. This clearly shows a large nonlocal signal at $n = \pm n_S$, corresponding to the moiré gaps at $D/\epsilon_0 = 0$ (Fig. 2c for $n = -n_S$ and Fig. 2e for $n = n_S$). In Fig. 2d, we plot the resistance at CNP for $D/\epsilon_0 = -0.3 \, \text{V nm}^{-1}$. We additionally plot the ohmic contribution to the nonlocal resistance due to stray current[39] in Fig. 2c–e. The calculated ohmic contribution (in "Methods"), being at least two orders of magnitude lower, cannot account for the large nonlocal resistance we observe at the gaps.

**Berry curvature spreading.** Now, we discuss an interesting difference in nonlocal resistance of TDBG compared to hBN-aligned MLG[28] or gapped BLG[29,30]. In a flat band system, the kinetic energy of the electrons is quenched. As the electrons slow down, with reduced Fermi velocity $v_F$, they start to see an enhanced effect of the other energy scales in the system,

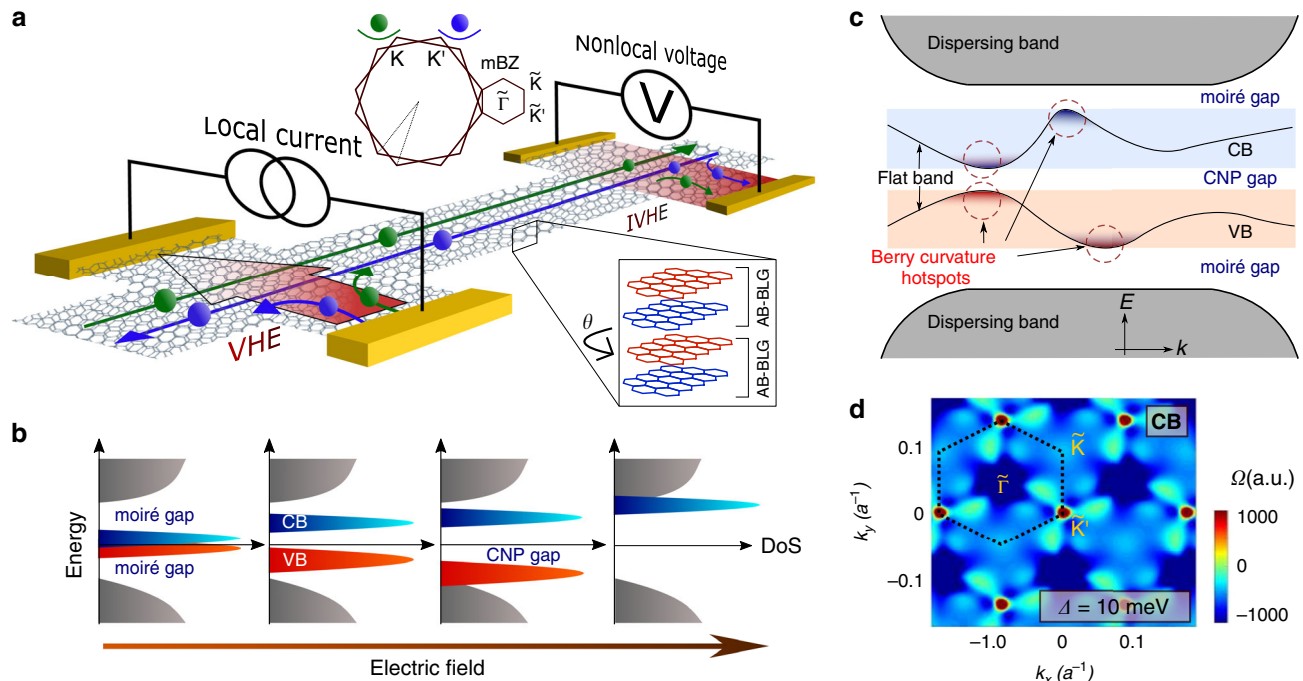

**Fig. 1 Nonlocal measurement scheme and electric field tunability in TDBG. a** Nonlocal measurement scheme of bulk valley current. A net valley current is generated by the charge current through the local probes in its transverse direction due to nonzero Berry curvature via valley Hall effect (VHE). In the nonlocal probes, the valley current generates a voltage drop by inverse VHE. **b** A schematic depicting the electric field tunable moiré bands in twisted double bilayer graphene (TDBG). Two low-energy flat bands—the conduction band (CB in blue) and the valence band (VB in red) are separated from high energy dispersing bands (gray) by two electric field tunable moiré gaps. CNP gap opens up between two flat bands as electric field is increased. **c** Schematic of a band structure of TDBG at a finite electric field with locations of the Berry curvature hotspots encircled. **d** A map of calculated Berry curvature $\Omega$ in **k**-space for K valley of the conduction flat band at a finite electric field corresponding to interlayer potential difference $\Delta = 10$ meV. Here $a$ is the lattice constant of graphene. This map shows the hotspots located at the symmetry points $\tilde{\Gamma}$, $\tilde{K}$, and $\tilde{K}'$ in the moiré Brillouin zone. The Berry curvature for K' valley has same magnitude but opposite sign.

for example, the $e$–$e$ interaction. In a similar way, smaller $v_F$ renormalizes the gap. The enhancement of the effective band gap results in the spreading of the Berry curvature hotspots. To quantitatively understand this effect, we consider the Berry curvature of a gapped ($2\Delta_g$) MLG with renormalized $v_F$ to incorporate the effect of band flatness, $\Omega(\mathbf{k}) = \frac{(\hbar v_F)^2 \Delta_g}{2[(v_F \hbar \mathbf{k})^2 + \Delta_g^2]^{3/2}}$. We find that the Berry curvature hotspot extends more in the **k**-space as $v_F$ is decreased (Supplementary Fig. 10 and Supplementary Note 8). As a result, $R_{NL}$ is appreciable over a large range of charge density around the gaps in TDBG. This is evident in Fig. 2c–e, as the nonlocal resistance peaks are broad in the charge density axis. On the other hand, in the earlier reported systems[28–30], nonlocal resistance falls more rapidly than the local resistance, as the charge density is tuned away from the gaps (comparison in Supplementary Fig. 11 and Supplementary Note 9). The spreading of Berry curvature is further supported by our finding that nonlocal resistance peaks have smaller normalized width for higher electric fields (see Supplementary Fig. 12 and Supplementary Note 10). This is because the bandwidth of the flat bands increases with the electric field, resulting in increase of $v_F$ (ref. [9]).

**Evidence of bulk valley transport**. We now proceed to understand the microscopic origin of the nonlocal signal. For diffusive transport of valley polarized electrons through the bulk, the nonlocal resistance $R_{NL}$ generated via VHE is given by[28]:

$$R_{NL} = \frac{1}{2}\left(\frac{\sigma_{xy}^{VH}}{\sigma_{xx}}\right)^2 \frac{W}{\sigma_{xx} l_v} \exp\left(-\frac{L}{l_v}\right). \quad (1)$$

Here, $\sigma_{xy}^{VH}$ is the valley Hall conductivity, $l_v$ indicates the valley diffusion length, with $L$ and $W$ being the length and the width of the Hall bar channel, respectively. This equation holds good when $\sigma_{xy}^{VH}/\sigma_{xx} \ll 1$, and results in a scaling relation between the local and the nonlocal resistance, $R_{NL} \propto R_L^3$ with $R_L = 1/\sigma_{xx}$.

To examine the scaling relation, we measure temperature dependence of the local and the nonlocal resistance for different $D$, as plotted in Fig. 3a, b, respectively, for the case of CNP (see Supplementary Fig. 13 for the case of $n = -n_S$). The local resistance shows Arrhenius activation behavior due to gaps in the system. The nonlocal resistance also follows activation behavior, but with higher gaps than the local resistance. In Fig. 3c, d, we plot the ratio of the nonlocal to the local gap as a function of electric field for $n = 0$ and $n = -n_S$, respectively. The inset of Fig. 3c, d shows the values of the activation gaps as a function of electric field. Although the individual gaps are tuned by the electric field, the ratio varies within 2.3–3.5. The ratio being close to 3 establishes $R_{NL} \propto R_L^3$, and hence supports bulk valley transport through Eq. (1). Also, this measurement reinforces our understanding that the contribution of $R_L$ in $R_{NL}$ is minimal.

Now, we closely examine the cubic scaling relation as a function of temperature (for scaling as a function of electric field, see Supplementary Fig. 7 and Supplementary Note 5). In Fig. 3e, f, we plot the nonlocal resistance against the local resistance in logarithmic scale, with temperature as a parameter. Figure 3e shows the case for $n = -n_S$, where the temperature varies from 10 to 75 K. The scaling remains cubic, with deviation at low $T$. This low temperature deviation from cubic scaling to being nearly independent of local resistance is consistent with the literature[29,30]. At low temperature, the system enters into a large

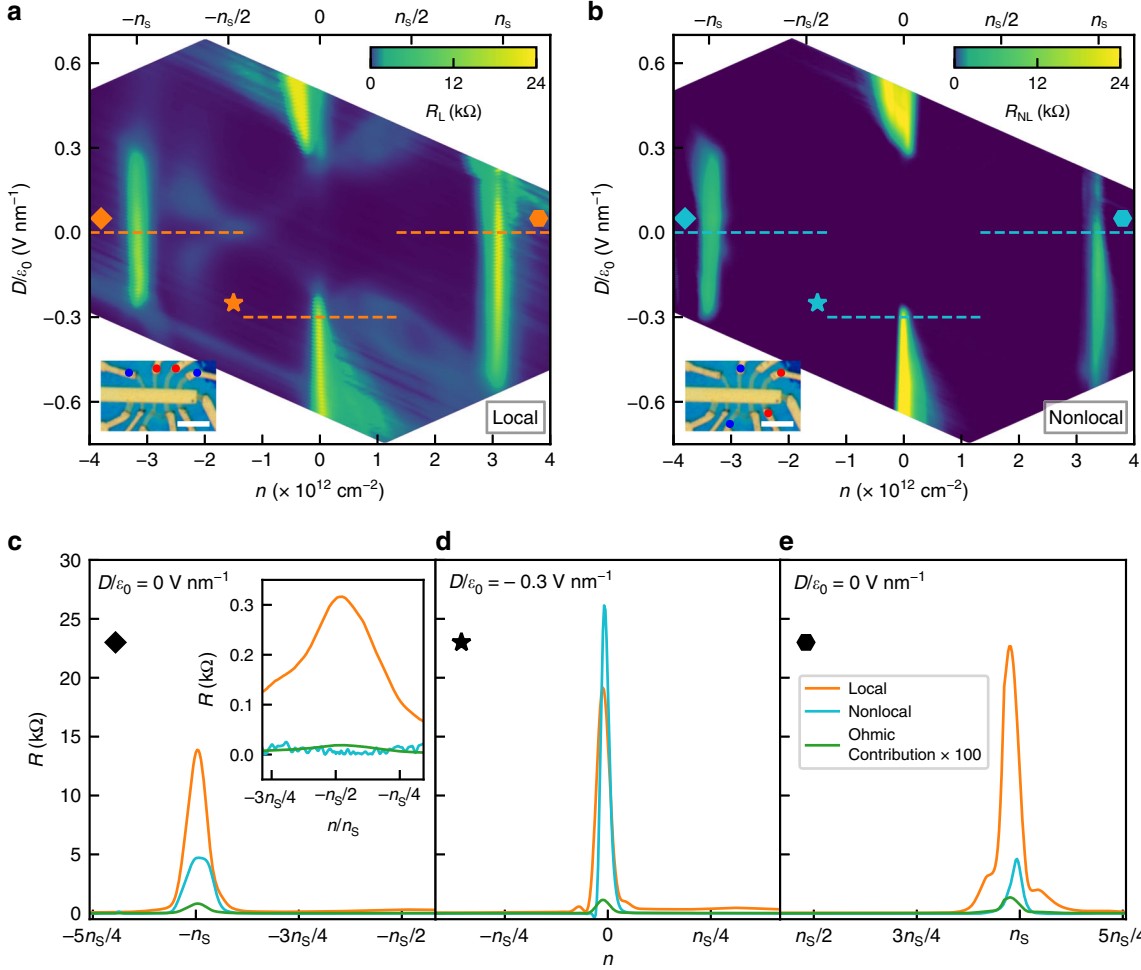

**Fig. 2 Local and nonlocal resistance in TDBG device with twist angle 1.18° at 1.5 K. a, b** Local and nonlocal resistance as a function of charge density and electric field. The dashed overlay line shows the location of line slices plotted in **c–e**. $n_S$ denotes the charge density to fill one moiré flat band. Insets show the micrographs of the device with voltage (current) terminals indicated by red (blue) dots. The scale bar corresponds to 5 μm. **c–e** Both local and nonlocal resistance, along with the calculated ohmic contribution, around moiré gaps at $n = -n_S$ (**c**), and $n = n_S$ (**e**) for the zero electric field and around the CNP gap at $n = 0$ (**d**) for a nonzero electric field. The nonlocal resistance is much larger than the ohmic contribution at the gaps. The inset in **c** zooms the resistance variation around $n = -n_S/2$ at $D/\epsilon_0 = 0$ V nm$^{-1}$, where nonlocal resistance is almost zero in spite of significant local resistance.

valley Hall angle regime, where the assumption $\sigma_{xy}^{VH}/\sigma_{xx} << 1$ is no longer valid[40].

The case where the system is at the charge neutrality is shown in Fig. 3f (the chosen range of temperature for showing scaling is shaded by blue in Fig. 3a, b). The scaling is cubic in intermediate temperature range, consistent with bulk valley transport, with departures at both ends. We note that $\sigma_{xy}^{VH}$ in Eq. (1) can have temperature dependence and decrease from its quantized value at elevated temperatures compared to the gap[29]. Such a phenomenon can explain the departure from cubic scaling at the high temperature end. The low temperature deviation at CNP is different from that in the case of $-n_S$, as a transition to higher power laws occurs. We note that for the case of nonlocal signal at CNP, the Fermi energy lies between two flat bands. Thus, at low temperatures, strong electron–electron correlations may give rise to edge states[41].

## Discussion

TDBG offers a unique platform since it provides electrical control over the flatness of Chern bands through $\nu_F$ and the band gap it hosts. Our study shows that one can further use this electrical control to induce bulk valley current and offers new opportunities

in valleytronics—via manipulating valley current by the tunable band gaps and the band flatness in twistronics. In particular, we show that the renormalized velocity in a flat band causes electrically controllable momentum spreading out of the Berry curvature hotspots. While we observe bulk valley current at elevated temperature, we cannot exclude the possibility that at low temperatures, the nonlocal response is additionally mediated by the edge modes associated with the VHE or other spontaneous quantum Hall effect[41,42] resulting from flat Chern bands[19].

The recent observation of anomalous Hall effect (AHE) and orbital magnetism in hBN-aligned TBG has been associated with the occupation of an excess valley- and spin-polarized Chern band by spontaneously breaking time-reversal symmetry[5,6]. The underlying topological structure of the bands plays an important role as opposite Chern numbers for two valleys preclude inter-valley coherence[20,21]. Our demonstration of valley current using nonlocal transport in TDBG as a result of opposite Berry curvature from two valleys complements this understanding. Our work provides strong evidence that VHE state, when the valley degeneracy is preserved, is indeed the parent state of the AHE state. We expect AHE state in TDBG as well, when the valley symmetry is broken. In addition, our work opens up new possibilities to explore chargeless valley transport in other moiré

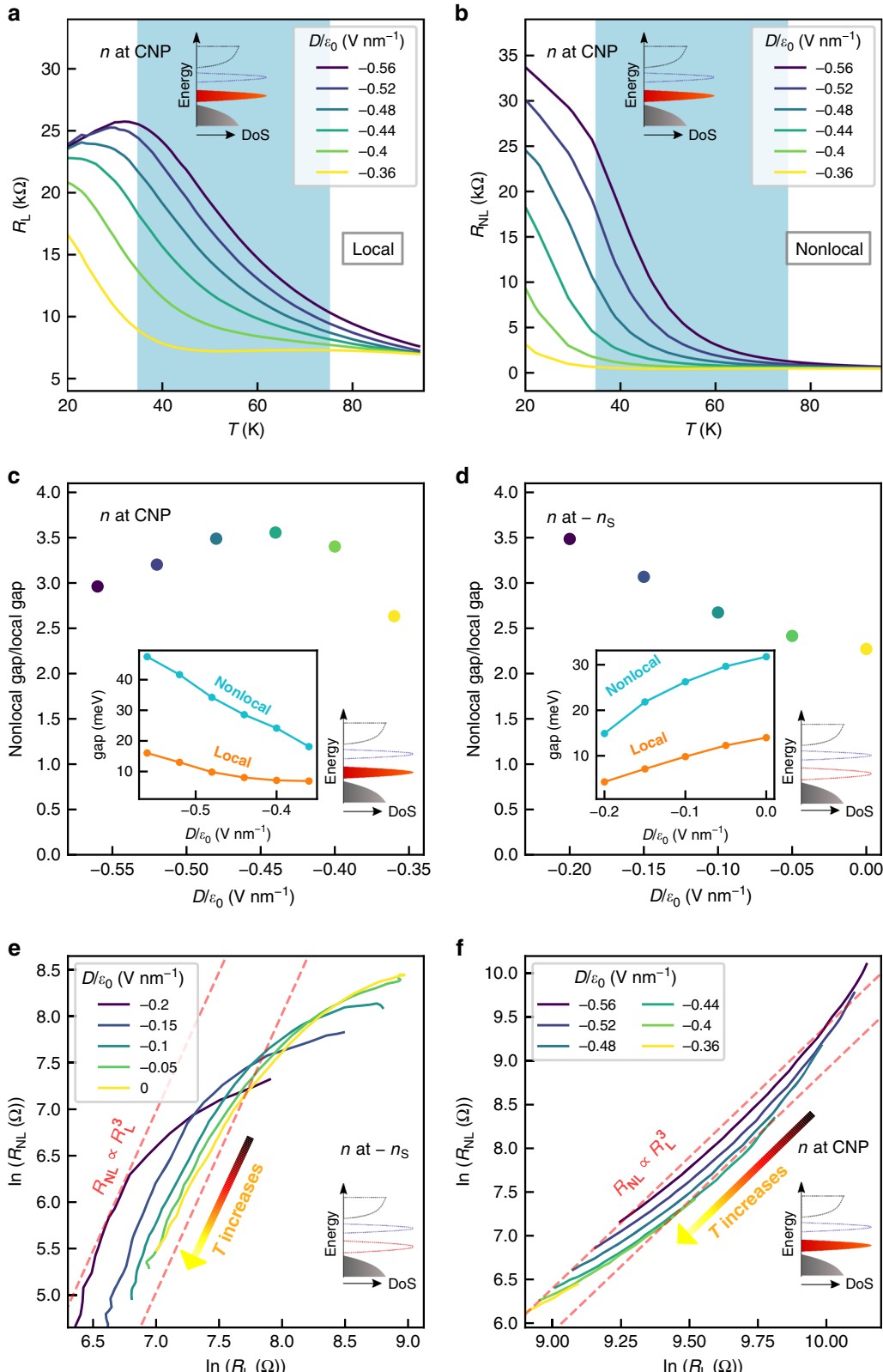

**Fig. 3 Temperature dependence and scaling of local and nonlocal resistance.** Inset of each panel shows schematic of band filling in that measurement. **a**, **b** Variation of local and nonlocal resistance with temperature at CNP for different electric fields. The region shaded with blue color is governed by Arrhenius activation and is the region used to show scaling relation in **f**. **c**, **d** Ratio of activation gaps for local and nonlocal resistance for CNP (**c**) and the moiré peak at $n = -n_S$ (**d**). The colors of the data points in **c** correspond to the same $D$ values as in **a**, **b**, **f**, and those in **d** are same as in **e**. The gaps are extracted by fitting Arrhenius activation equation to $R$ vs $T$ curves in the temperature range within 15–85 K. Insets show the variation of gaps with electric field. The ratio, being close to 3, shows that $R_{NL} \propto R_L^3$, and thus strongly supports bulk valley transport. **e**, **f** The scaling of nonlocal resistance with local resistance at different electric fields for $n = -n_S$ (**e**) and CNP (**f**). Here, temperature is used as a parameter to show the scaling.

systems like trilayer graphene aligned with hBN, twisted trilayer graphene, and other twisted transition metal dichalcogenides having topological Chern bands.

## Methods

**Device fabrication.** Our dual-gated devices are made of hBN/TDBG/hBN stacks on $SiO_2$ (~280 nm)/$Si^{++}$ substrate. To make the stacks, we exfoliate graphene and choose suitable bilayer graphene flakes based on optical contrast, and then confirm the layer number by Raman spectroscopy. The suitable hBN flakes are selected based on color, and we confirm the thickness by AFM after the stack is completed. Bilayer graphene flakes are sliced into two halves, using a tapered optical fiber scalpel prepared with an optical fiber splicer[43]. Subsequently, the flakes are assembled using the standard poly(propylene) carbonate-based dry transfer method[44]. The twist angle is introduced by rotating the bottom stage during the pick up of the second half of the graphene. Subsequently, we define the geometry of the devices by e-beam lithography, followed by $CHF_3 + O_2$ plasma etching. One-dimensional edge contacts to the graphene are made by etching the stack and depositing Cr/Pd/Au. The top gate is made by depositing Cr/Au.

**Measurements.** We fabricate and measure nonlocal transport in multiple devices. The dual-gated structure using a metal top gate and highly doped silicon back gate enables us to have independent control of both the charge density ($n$), and the perpendicular electric displacement field ($D$) given by $n = (C_{TG}V_{TG} + C_{BG}V_{BG})/e$ and $D = (C_{TG}V_{TG} - C_{BG}V_{BG})/2$, where $C_{TG}$ and $C_{BG}$ are the capacitance per unit area of the top and the back gate, respectively, and $e$ is the charge of an electron. All the data reported in the main manuscript are measured using device 1 with twist angle 1.18°. We present data from another device with twist angle 1.24° in the Supplementary Information (see Supplementary Fig. 8 and Supplementary Note 6).

The transport measurements reported in the main manuscript and the Supplementary Information are conducted using a low frequency (~17 Hz) lock-in technique by sending a current ~10 nA and measuring the voltage after amplifying, using SR560 preamplifier or preamplifier model 1021 by DL instruments, Ithaca. In addition, we perform dc measurements to verify that the measured nonlocal signal is repeatable and independent of the measurement scheme (Supplementary Fig. 4 and Supplementary Note 2). We further checked that the nonlocal signal is consistent with reciprocity (Supplementary Fig. 6 and Supplementary Note 4). The ohmic contribution to the nonlocal resistance due to stray current, as plotted in Fig. 2c, has been calculated by using the van der Pauw formula, $R_{NL}^{ohm} = (\rho_{xx}/\pi) \times \exp(-\pi L/W)$[29]. Here, $L$ and $W$ are the length and width of the conduction channel, respectively. For the device presented in the main text, we choose $L = 4\,\mu m$ and $W = 2\,\mu m$ for performing the nonlocal measurements. This contribution decays exponentially along the length of the conduction channel. For the local measurements, we use the four-probe method and choose both $L$ and $W$ to be $2\,\mu m$ from the same device, allowing us to use the relation $R_L = 1/\sigma_{xx}$ in the main text.

**Determining the twist angle.** For determining the twist angle, we first determine $n_S$, the charge density to fill one moiré flat band. On the plot of resistance as a function of charge density ($n$) and the perpendicular electric field ($D/\epsilon_0$), the difference in the location of the two resistance peaks in the charge density axis (the resistance peaks corresponds to two moiré gaps at $\pm n_S$ at $D = 0$), gives $2n_S$. The twist angle $\theta$ is then calculated from $n_S$ using the relation $n_S = 8\theta^2/(\sqrt{3}a^2)$, where $a = 0.246$ nm is the lattice constant of graphene. From Fig. 2a, we find $n_S = 3.2 \times 10^{12}$ cm$^{-2}$, giving a twist angle of 1.18° for device 1 in the main manuscript.

## Data availability

The data used to produce Figs. 2 and 3 in the main text are available in Zenodo with the identifier https://doi.org/10.5281/zenodo.3960483 (ref. [45]). Additional data related to this study are available from the corresponding authors upon reasonable request.

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

## Acknowledgements

We thank Allan MacDonald, Justin Song, Rajdeep Sensarma, Vibhor Singh, Sajal Dhara, and Biswajit Datta for helpful discussions and comments. We acknowledge Nanomission grant SR/NM/NS-45/2016 and DST SUPRA SPR/2019/001247 grant along with Department of Atomic Energy of Government of India 12-R&D-TFR-5.10-0100 and ITC-PAC contract no. FA520920P0093 for support. Preparation of hBN single crystals is supported by the Elemental Strategy Initiative conducted by the MEXT, Japan, and JSPS KAKENHI grant no. JP15K21722. This work is supported by the Korean NRF for B.L.C. through Basic Science Research Program of the National Research Foundation of Korea (NRF) funded by the Ministry of Education grant no. 2018R1A6A1A06024977 and grant no. NRF-2020R1A2C3009142, and for J.J. through Samsung Science and Technology Foundation under project no. SSTF-BA1802-06.

## Author contributions

P.C.A., S.S., R.S.S.K., and L.D.V.S. fabricated the devices. P.C.A. and S.S. did the measurements and analyzed the data. B.L.C. and J.J. did the theoretical calculation. K.W. and T.T. grew the hBN crystals. P.C.A, S.S., and M.M.D. wrote the manuscript with inputs from everyone. M.M.D. supervised the project.

## Competing interests

The authors declare no competing interests.
