## [Peer Review File · Nature Communications]

REVIEWER COMMENTS

Reviewer #1 (Remarks to the Author):

In this manuscript, the authors report some interesting results on the Berry-curvature induced nonlocal transport in twisted double AB bi-layer graphene. Previously, similar experiments have been reported in simple AB bi-layer graphene (refs. 23 and 24). Recently, the nonlocal transport properties have also been reported in simple twisted bilayer graphene (<https://arxiv.org/abs/1903.07950>). Overall the results reported in this manuscript enrich the topological transport properties in graphene materials. I feel that the manuscript can be published after addressing the following comments.

1. They should compare their results with the results reported in <https://arxiv.org/abs/1903.07950>, since these two manuscripts are closely related.
2. BLG/BLG has much smaller gaps compared to all other systems such as simple A-B bi-layer under vertical field and simple twisted bilayer. What is the impact of the smaller bandgap on nonlocal transport if compared with previous A-B bi-layer and twisted bilayer?
3. It is not clear how they determine the twist angle, and how they determine the Chern numbers and their signs experimentally.

Reviewer #2 (Remarks to the Author):

The manuscript by S. Sinha et al. reports the first experimental observations of bulk valley current in twisted double bilayer graphene (TDBG), as manifested by the nonlocal transport measurements. The central findings of this work are interesting and will make impact on the exotic physics of twisted graphene layers. However, for TDBG, this manuscript does not represent a major breakthrough since the valley current can be detected only when the inversion symmetry is broken (by electric field, substrate screening, ect.). Such valley current has been experimentally reported in earlier studies in bilayer graphene (Nat. Phys. 2015, 11, 1027–1031 and Nat. Phys. 2015, 11, 1032-1036), also the bilayer graphene can be realized as a gate-tunable valley splitter (Science 2018, 362, 1149–1152.). Based on the overall novelty of this manuscript, I do not recommend its publication in Nature Communications, at least in the present form. Perhaps, this manuscript is acceptable with major revisions after the authors clarify the following questions:

- 1) In the low-angle TDBG device, the authors did not observe the correlated states at half fillings (ref. Shen et al, Nat. Phys. 2020). They need to explain why.
- 2) In Figure 2, the measured nonlocal resistance (up to 25k ohm) is much larger than the values in previous reports of bilayer graphene (600 ohms, Nat. Phys. 2015, 11, 1027–1031.). The authors need to explain the discrepancy. Is this phenomenon relevant to the flat band?
- 3) The nonlocal resistance peak is broad in the charge density axis, which is corresponding to the carrier density (Figure 2c), and the authors claim that this could be the result of extended Berry curvature hotspots in k-space as Fermi velocity is decreased. This explanation remains to be verified by providing more experimental data with different twisted angles, especially the large twisted angle devices, in which the band is not flat and the Fermi velocity is large enough.

4) Most importantly, the main finding of this work, according to the authors, is that they observed the VHE state in TDBG, which highlights the importance as it is a precursor of AHE state. However, the VHE state can be easily observed in biased bilayer graphene. Thus it is reasonable that VHE state can also exist in TDBG due to the broken inversion symmetry. The authors should further elaborate the central findings of this work to justify the novelty and significance to be considered for publication in Nature Communications.

Reviewer #3 (Remarks to the Author):

The authors report the nonlocal transport measurement in the twisted double bilayer graphene. They found significant nonlocal resistances in the charge-neutrality gap and in the moire gaps, and attribute them to the valley Hall effect due to broken inversion symmetry. The authors also confirmed the qualitative behavior such that the nonlocal resistance is proportional to the cube of the local resistance.

The paper is the first measurement of the non-local transport in the flat band systems, and potentially acceptable for publication. The experimental data looks carefully analyzed to conclude that the non-local resistance is due to the Berry curvature effect, but still I find there are several issues to be considered. I request the authors to address the following problems:

--- In principle, the magnitude of valley Hall conductivity σ_{xy}^{VH} can be estimated using Eq. (1) with measured R_{NL} and R_L , estimated I_v (Sec. VII in SI) and L and W from the sample geometry. Do we get σ_{xy}^{VH} roughly in the order of e^2/h (theoretical calculation), or some different magnitudes (then why)?

--- In the ideal system, the σ_{xy}^{VH} is $2e^2/h$ for the CNP gap and 0 for the moire gaps. I wonder why we see finite non-local signals in the moire gaps in the experiment. I imagine the moire gaps could be smeared by the temperature (or some other broadening effects) like in Fig. S3, but then non-local signal should go to zero in the low temperature limit. (The reader might want to see the T-dependence plot like Fig 3ab also for the moire gaps.) In Fig. 2 ($T=1.5K$), we have a significant nonlocal resistance in the moire gaps, even though the peak of R_L is very sharp (=the gap looks well formed).

--- Concerning the statement in page 7:
> We note that the nonlocal signal at CNP originates
> due to Berry curvature hotspots located at the edge of flat bands
> (See Supplementary Sec. I for σ_{VH} at CNP in TDBG).

I feel emphasizing the "hot spots" may be confusing, because the nonlocal signal at CNP is related to the quantized in-gap valley Hall conductivity ($\sigma_{VH}=2e^2/h$) as plotted in Fig.S3, which is the integral of the Berry curvature all over the states below the gap.

Having a strong local Berry curvature at a particular k-point doesn't seem essential for observing non-local signals.

(it is just related to the derivative of σ_{VH} in energy)

Indeed, the author states that R_{NL} is appreciable over larger range of charge density when the hotspot is more spread in k-space.

Minor points:

--- Fig. 1d: provide Δ .

--- Fig. 3c and 3d: provide temperature.

We thank the reviewers for their helpful comments; they have contributed to making the manuscript stronger and we are grateful.

We have also now included some of our raw data to be freely accessible via <https://doi.org/10.5281/zenodo.3960483>.

Below, we address the comments/questions of the reviewers:

(Reviewers' comments/questions are in the *blue colored italic font* and our responses are in the black colored font)

Response to Reviewer #1:

In this manuscript, the authors report some interesting results on the Berry-curvature induced nonlocal transport in twisted double AB bi-layer graphene. Previously, similar experiments have been reported in simple AB bi-layer graphene (refs. 23 and 24). Recently, the nonlocal transport properties have also been reported in simple twisted bilayer graphene (<https://arxiv.org/abs/1903.07950>). Overall the results reported in this manuscript enrich the topological transport properties in graphene materials. I feel that the manuscript can be published after addressing the following comments.

We thank the reviewer for the constructive and positive review. Below we address all the comments by the reviewer.

Comment #1: *1. They should compare their results with the results reported in <https://arxiv.org/abs/1903.07950>, since these two manuscripts are closely related.*

Reply1: We thank the reviewer for pointing to this study. While both the works report nonlocal transport in flat band systems, there are a few key differences as we discuss below.

- In our work, the nonlocal resistance observed in twisted double bilayer graphene (TDBG) originates due to nonzero Berry curvature. This is consistent with other broken inversion symmetry systems explored in the literature such as hBN aligned monolayer graphene, bilayer graphene under electric field, MoS₂, etc. However, as reported in [arXiv:1903.07950](https://arxiv.org/abs/1903.07950), Berry curvature is zero due to symmetry in twisted bilayer graphene (TBG). The authors attribute the origin of the nonlocal resistance to “high dimensional band topology”-- an aspect that is not clear to us, and it seems to be twisted bi-layer specific physics.
- The scaling relation $R_{NL} \propto R_L^3$ is consistent with bulk valley transport which originates by valley Hall effect due to nonzero Berry curvature in TDBG. However for TBG in [arXiv:1903.07950](https://arxiv.org/abs/1903.07950), the scaling relation is explained by transport due to “quasi-one-dimensional diffusion channels” formed by “the spreading of edge states into the bulk”, which we do not fully understand.

- The nonlocal transport in TDBG is highly tunable by the perpendicular electric field due to the tunability of the band structure itself by the electric field. In contrast, the band structure of TBG is largely immune to the electric field. In [arXiv:1903.07950](https://arxiv.org/abs/1903.07950), the authors attribute the weak modulation of nonlocal resistance by the electric field to the formation of “network of AB-BA domain wall modes”. In TDBG there are no such networks.

Action1: We have included the work [arXiv:1903.07950](https://arxiv.org/abs/1903.07950) in the references of our revised manuscript. Additionally, to orient the readers we added these sentences at the end of the second paragraph in the revised main manuscript: “In a recent study the nonlocal resistance reported in twisted bilayer graphene has been attributed to high dimensional topology as the symmetry enforces Berry curvature to be zero. However, the Berry curvature mediated bulk valley transport remains to be explored in a flat band system.”

Comment #2: *2. BLG/BLG has much smaller gaps compared to all other systems such as simple A-B bi-layer under vertical field and simple twisted bilayer. What is the impact of the smaller bandgap on nonlocal transport if compared with previous A-B bi-layer and twisted bilayer?*

Reply2: We thank the reviewer for this interesting observation. Using the equation (1) of the main manuscript, the impact of the smaller bandgap on nonlocal transport can be understood from the dependency of σ_{xy}^{VH} and $R_L \propto 1/\sigma_{xx}$ on the bandgap,

$$R_{NL} = \frac{1}{2} \left(\frac{\sigma_{xy}^{VH}}{\sigma_{xx}} \right)^2 \frac{W}{\sigma_{xx} l_v} \exp\left(-\frac{L}{l_v}\right).$$

- Near the gap, σ_{xy}^{VH} is constant. But as the temperature is increased R_L decays much rapidly due to activated transport governed by smaller bandgap. This is reflected in R_{NL} accordingly, as seen in Fig. 3b of the main manuscript, where we see R_{NL} decays more rapidly compared to AB-bilayer graphene or TBG.
- As we discuss in the Supplementary Note 8, away from the gap σ_{xy}^{VH} decays governed by the parameter $\delta = \Delta_g / \hbar v_F$. For smaller bandgap Δ_g , σ_{xy}^{VH} decays more rapidly and hence nonlocal resistance peaks should have narrow width. However, this is compensated by renormalized Fermi velocity v_F resulting in Berry curvature spreading, a novel aspect of valley transport in the flat band system as explored in our work.
- The magnitude of the Berry curvature at the gap increases as the bandgap decreases: $\Omega \sim 1/\Delta_g^2$. However, how that impacts the nonlocal resistance is complex to predict a trend as σ_{xx} in the activated region depends on the gap.

Comment #3: *3. It is not clear how they determine the twist angle, and how they determine the Chern numbers and their signs experimentally.*

Reply3: We regret that the description given in the ‘Methods’ section of the main manuscript about how we determine the twist angle was not clear. In line with the common practice, we infer the twist angle θ using the relation, $n_S = 8\theta^2/(\sqrt{3}a^2)$, where n_S is the charge density to fill one moiré flat band and $a = 0.246$ nm is the lattice constant of graphene.

As shown in Fig. R1 (reproduced from Fig. 2a of the main manuscript), we determine n_S by locating the two resistance peaks corresponding to two moiré gaps at $\pm n_S$ on the plot of resistance as a function of charge density n , and the electric field D/ϵ_0 . The charge density n , and the electric field D is determined from the top and the back gate voltages V_{TG} and V_{BG} , using the equations $n = (C_{TG}V_{TG} + C_{BG}V_{BG})/e$, and $D = (C_{TG}V_{TG} - C_{BG}V_{BG})/2$. Here, the gate capacitances C_{TG} and C_{BG} are known experimentally.

For example, from Fig. R1 for the case of device 1, we find $n_S = 3.2 \times 10^{12} \text{ cm}^{-2}$; and using the relation $n_S = 8\theta^2/(\sqrt{3}a^2)$ we find the twist angle to be 1.18° .

Fig. R1: Determining twist angle: Color scale plot of resistance as a function charge density n and the electric field D/ϵ_0 . The two moiré peaks are observed at $n = \pm n_S$. Hence the difference between the locations of two moiré peaks $2n_S$ is used to determine twist angle using the equation, $n_S = 8\theta^2/(\sqrt{3}a^2)$.

While our theoretical calculation (Supplementary Fig. 2) shows that the bands in TDBG have non-zero valley Chern numbers, we do not experimentally infer any information about the Chern number. As the theoretical calculation predicts, TDBG may show the anomalous Hall effect

(Jianpeng Liu et al. *Phys. Rev. X* **9**, 031021 (2019)) when time-reversal symmetry is broken. In such an experiment one can measure the Chern number by measuring the quantized value of anomalous Hall resistance R_{xy} , since $\sigma_{xy} = Ce^2/h$ where C is the Chern number. This is beyond the scope of our present study as time-reversal symmetry is preserved in our experiments.

Action3: In the 'Methods' section of the revised manuscript we have improved the description of how we determine the twist angle.

Response to Reviewer #2:

The manuscript by S. Sinha et al. reports the first experimental observations of bulk valley current in twisted double bilayer graphene (TDBG), as manifested by the nonlocal transport measurements. The central findings of this work are interesting and will make impact on the exotic physics of twisted graphene layers. However, for TDBG, this manuscript does not represent a major breakthrough since the valley current can be detected only when the inversion symmetry is broken (by electric field, substrate screening, ect.). Such valley current has been experimentally reported in earlier studies in bilayer graphene (Nat. Phys. 2015, 11, 1027–1031 and Nat. Phys. 2015, 11, 1032-1036), also the bilayer graphene can be realized as a gate-tunable valley splitter (Science 2018, 362, 1149–1152.). Based the overall novelty of this manuscript, I do not recommend its publication in Nature Communications, at least in the present form. Perhaps, this manuscript is acceptable with major revisions after the authors clarify the following questions:

We thank the reviewer for the careful and constructive review. We believe that establishing twisted graphene systems as a possible valleytronics platform is a key achievement since twistrionics offers a new experimental band engineering control through the twist angle, which was absent in earlier reported systems. We thank the reviewer for pointing out the important work in *Science* 2018, 362, 1149–1152 that we have cited in the revised manuscript. Below we address all the concerns raised by the reviewer.

Comment #1: *1) In the low-angle TDBG device, the authors did not observe the correlated states at half fillings (ref. Shen et al, Nat. Phys. 2020). They need to explain why.*

Reply1: We appreciate the reviewer for this important question. At first, we would like to mention that our major focus is to investigate the bulk valley transport which occurs at a higher temperature. At such temperature range, the effect of correlation can be ignored as understood by the low values of correlated gaps. However, below we address the reviewer's question.

- As seen in Fig. R2, we do see enhanced correlated gaps as we increase the parallel magnetic field in consistence with the literature, while the absence of prominent resistance peak at $n_s/2$ in the zero magnetic field may indicate a very low value of the correlated gap.

Fig. R2: Enhanced correlation induced gaps at $D/\epsilon_0 = -0.4 \text{ V/nm}$ with the in-plane magnetic field.

- In TDBG, correlated states are most prominent near the twist angle of 1.3° to 1.4° , as it is required for the moiré flat bands to be well separated from the higher bands in addition to the low bandwidth [*Phys. Rev. Lett.* **123**, 197702 (2019)]. This is further supported by a recent experimental work in P. Rickhaus, K. Ensslin et al. (Fig. S2c), where they explore a TDBG device with a twist angle of 1.2° (close to 1.18° of our device).
- We would like to mention the calculation published by J. Y. Lee, A. Vishwanath et al. *Nature Communications* **10**, 5333 (2019) as shown in Fig. R3, which identifies the parameter space for the isolated conduction flat band in TDBG. The twist angle of 1.18° in our device (marked by a solid red line) lies in the vicinity of the optimal parameter space justifying weak correlated gaps we observed.

[Redacted]

Fig. R3: Isolation region of the first conduction band in the twist angle-electric field parameter space. The color scale indicates the bandwidth (in meV) of the first conduction band. White color represents the parameter space when the flat bands are not isolated from the higher bands. The solid red line indicates

the angle corresponding to our 1.18° TDBG device [reproduced from J.Y. Lee, A. Vishwanath et al., Nature Communications 10, 5333 (2019)].

Comment #2: *2) In Figure 2, the measured nonlocal resistance (up to 25k ohm) is much larger than the values in previous reports of bilayer graphene (600 ohms, *Nat. Phys.* 2015, 11, 1027–1031.). The authors need to explain the discrepancy. Is this phenomenon relevant to the flat band?*

Reply2: We thank the reviewer for this observation. However, from the following points we infer that the values of R_{NL} observed in our experiment is consistent with *Nat. Phys.* 2015, 11, 1027–1031.

- R_{NL} reported in Fig. 2 in our manuscript is measured at 1.5 K, whereas R_{NL} reported in *Nat. Phys.* 2015, 11, 1027–1031 is at 70 K. Since, R_{NL} falls rapidly with increasing temperature governed by charge carrier activation, we consider R_{NL} at 70 K from the Fig. 3b in the main manuscript for more meaningful comparison. In our case R_{NL} at 70 K varies from 1 k Ω to 3 k Ω for different values of electric field, thus within one order of magnitude of that reported in *Nat. Phys.* 2015, 11, 1027–1031.
- From equation (1) in the main manuscript, the measured nonlocal resistance falls exponentially with the distance l between local and nonlocal probes, $R_{NL} \propto e^{-ll_v}$. Here l_v is the valley diffusion length. While $l = 5 \mu\text{m}$ in the case of the referred paper, in our case $l = 4 \mu\text{m}$, which can justify the larger magnitude we observe.

Finally, we believe that the role of the flat bands will be more relevant at low temperature as the underlying valley Chern bands in TDBG can support edge states resulting in high nonlocal signals. While this is not the focus of our present study, we hope our experiment which explores bulk valley transport at higher temperature will motivate further study in that direction.

Comment #3: *3) The nonlocal resistance peak is broad in the charge density axis, which is corresponding to the carrier density (Figure 2c), and the authors claim that this could be the result of extended Berry curvature hotspots in k-space as Fermi velocity is decreased. This explanation remains to be verified by providing more experimental data with different twisted angles, especially the large twisted angle devices, in which the band is not flat and the Fermi velocity is large enough.*

Reply3: We appreciate the reviewer's suggestion to explore large twist angle devices. We had directly compared our TDBG with a system without flat bands (graphene with hBN superlattice) in Supplementary Fig. 11 and Supplementary Note 9 to show that this Berry curvature spreading is intrinsic to systems with flat bands. However, as the reviewer suggests, we discuss more experimental results and analysis in the following.

In Fig. R4, we show additional measurements in a TDBG device with a larger twist angle of 2.05°. However, we find that the band structure (and thus electronic transport) in TDBG can modify quite substantially with twist angle, thus making it hard to establish Berry curvature

spreading systematically as a function of twist angle. For example, the local resistance (Fig. R4a) at the charge neutrality point (CNP), $n=0$, first decreases and then increases as the gap is closed and reopened with an increasing electric field (consistent with earlier studies as in Fig. S7 of X. Liu, P. Kim et al. arXiv:1903.08130). Recent studies (Rickhaus, Ensslin et al. arXiv:2005.05373) show that the re-emerged gap is correlation induced and not a single-particle gap that we are interested in.

Fig. R4: Data from a higher twist angle (2.05°) TDBG device: (a) Local resistance showing gap closing and reopening at CNP. (b) Nonlocal resistance (plotted after normalizing it with the maximum nonlocal resistance in the dataset) from the same device.

Below we provide a more conclusive experimental support where we tune the Fermi velocity by changing the electric field within the same TDBG device having a fixed twist angle.

- The analysis we describe now is based on the fact that as the bandwidth reduces, that is the band becomes flatter, the Fermi velocity reduces. As seen in the inset of Fig. R5, by increasing the magnitude of perpendicular electric field (D) in TDBG the bandwidth of the flat bands is increased and thus the Fermi velocity is increased [Chebrolu et al. Phys. Rev. B **99**, 235417 (2019), J.Y. Lee, A. Vishwanath et al., Nature Communications **10**, 5333 (2019)]. We have plotted the nonlocal resistance in the 1.18° TDBG as a function of charge density across the CNP for three different electric fields in Fig. R5. We find that as we increase the electric field, the width of the nonlocal resistance peak becomes narrower thus confirming that lower Fermi velocity at low electric fields results in a wider nonlocal resistance peak.

[Redacted]

Fig. R5. Variation of nonlocal resistance (R_{NL}) with charge density (n) in TBDG (twist angle of 1.18°). The three line slices are plotted after normalizing with respect to the individual maximum resistance at the peak (R_{NL}^{peak}) to compare the FWHM. Increasing the bandwidth with $|D|$ increases the renormalized Fermi velocity of the flat band, decreasing the spread of R_{NL} in the charge density axis. Inset: Variation of the bandwidth of the flat band in the same D range [Adak et al., Phys. Rev. B, **101**, 125428 (2020)].

- We systematically extract the nonlocal FWHM and plot this variation with the electric field for two devices with twist angles 1.18° (Fig. R6a) and 1.24° (Fig. R6b). We find that this feature of decrease in FWHM of R_{NL} with increase in bandwidth of flat bands (by increasing $|D|$), and thus increase in Fermi velocity, is repeatable. This reinforces our understanding of the physics of Berry curvature spreading due to flat bands -- the spreading being related to the Fermi velocity.

Fig. R6. Variation of the FWHM of the nonlocal resistance peaks with the perpendicular electric field at CNP in TBDG with a twist angle of 1.18° (a) and 1.24° (b). The data points in (a) are extracted at $T=9.5$ K

(three of the nonlocal resistance curves have been shown in Fig. R5). The data points in (b) are extracted from device-2 (Supplementary Fig. 8).

Action3: We have included the details in Supplementary Fig. 12 and Supplementary Note 10 of the revised Supplementary Information as additional support to our explanation of Berry curvature spreading in a flat band system.

Comment #4: *4) Most importantly, the main finding of this work, according to the authors, is that they observed the VHE state in TDBG, which highlights the importance as it is a precursor of AHE state. However, the VHE state can be easily observed in biased bilayer graphene. Thus it is reasonable that VHE state can also exist in TDBG due to the broken inversion symmetry. The authors should further elaborate the central findings of this work to justify the novelty and significance to be considered for publication in Nature Communications.*

Reply4: We thank the reviewer for asking us to state the main findings of the paper. We mention them one by one:

- Like TDBG, the VHE state in bilayer graphene at CNP is observed due to broken inversion symmetry under a finite electric field. However, TDBG is very distinct from the bilayer graphene as the **moiré periodicity** in TDBG plays an important role to open up secondary gaps, thus decoupling the two K and K' valleys (J. Song et al. PNAS 112 (35), 10879-10883 (2015)) and **isolating** the flat bands. Isolated valley Chern bands together with electronic correlations due to band flatness make twisted graphene systems a very rich and novel topological platform. Currently, understanding these topological aspects is a major focus as evidenced by a number of recent experimental works as we write this response (such as X. Lu, D. K. Efetov et al., arXiv:2006.13963, and K. P. Nuckolla, A. Yazdani et al. arXiv:2007.03810 in TBG; and G. W. Burg, E. Tutuc et al. arXiv:2006.14000 in TDBG). In the present study, we experimentally explore the topological aspect in TDBG for the first time.
- Our work has far-reaching significance considering the recent observations of AHE state in hBN aligned TBG. Anomalous Hall effect has been observed in different condensed matter systems under different settings such as large spin-orbit interaction, magnetic dopants, etc. However in recent theoretical works, the AHE state in twisted graphene systems has been associated with the spontaneous polarization in spin and valley space governed by the underlying topological structure of the bands as opposite Chern numbers for two valleys preclude uniform intervalley coherence (N. Bultinck, M. P. Zaletel et al., PRL 124, 166601 (2020)). Our experimental observation of valley current supported by isolated valley Chern bands with opposite Berry curvature for K and K' valleys directly complements the present theoretical understanding. While we expect AHE also in TDBG, our work thus probes the VHE state as the precursor of the AHE state before the valley degeneracy is lifted.
- The valley current was first observed in a moiré system of hBN-aligned-graphene (Gorbachev et al., Science 346, 448 (2010)) which can host isolated topological bands. The observation of valley current in BLG offers the electric field as an important control

knob. TDBG enables realizing both these functionalities in a single system as we explore bulk transport of chargeless valley current for the first time in **any** twisted graphene system. This may be an important step forward in valley-twistronics applications.

- In addition to electric field tunability, the extreme flatness of the bands in TDBG opens up further opportunities. One such example is already explored in our work as we observe that the Berry curvature hotspot is not merely localized near the band edge, rather spread more over the k-space.

Action4: We have modified the third paragraph of the 'Introduction' section where we introduce the TDBG system and the last paragraph of the 'Discussion' section at the end of the revised manuscript to bring out the novel aspects of our work more clearly.

Response to Reviewer #3:

The authors report the nonlocal transport measurement in the twisted double bilayer graphene. They found significant nonlocal resistances in the charge-neutrality gap and in the moire gaps, and attribute them to the valley Hall effect due to broken inversion symmetry. The authors also confirmed the qualitative behavior such that the nonlocal resistance is proportional to the cube of the local resistance.

The paper is the first measurement of the non-local transport in the flat band systems, and potentially acceptable for publication. The experimental data looks carefully analyzed to conclude that the non-local resistance is due to the Berry curvature effect, but still I find there are several issues to be considered.

I request the authors to address the following problems:

We thank the reviewer for the constructive review and appreciation of our work. Below we address all the concerns raised by the reviewer.

Comment #1: *--- In principle, the magnitude of valley Hall conductivity σ_{xy}^{VH} can be estimated using Eq. (1) with measured R_{NL} and R_L , estimated I_v (Sec. VII in SI) and L and W from the sample geometry.*

Do we get σ_{xy}^{VH} roughly in the order of e^2/h (theoretical calculation), or some different magnitudes (then why)?

Reply1: The reviewer is correct that in principle one can calculate σ_{xy}^{VH} using the equation (1) of the main manuscript. For example, in Fig. R7 below, we have calculated the value of σ_{xy}^{VH} for different values of the electric field using $l = 4 \mu\text{m}$, $w = 2 \mu\text{m}$, and the estimated value of $l_v = 1.15 \mu\text{m}$ (Supplementary Fig. 9b and Supplementary Note 7) at the temperature of 35 K. We used the values of R_L and R_{NL} at the same temperature of 35 K as in the Fig. 3a and 3b in the main manuscript. The values of σ_{xy}^{VH} are similar to the theoretically expected value of $\sigma_{xy}^{VH} = \sigma_{xy}^K - \sigma_{xy}^{K'} = 4 e^2/h$.

Fig. R7: Extracted values of σ_{xy}^{VH} for different values of the electric field at 35 K using equation (1) of the main manuscript.

Comment #2: --- In the ideal system, the σ_{xy}^{VH} is $2e^2/h$ for the CNP gap and 0 for the moire gaps. I wonder why we see finite non-local signals in the moire gaps in the experiment. I imagine the moire gaps could be smeared by the temperature (or some other broadening effects) like in Fig. S3, but then non-local signal should go to zero in the low temperature limit. (The reader might want to see the T-dependence plot like Fig 3ab also for the moire gaps.) In Fig. 2 ($T=1.5K$), we have a significant nonlocal resistance in the moire gaps, even though the peak of R_L is very sharp (=the gap looks well formed).

Reply2: We thank the reviewer for the careful observation. A theoretical calculation of σ_{xy}^{VH} cannot capture all the detailed evolution of R_{NL} , since R_{NL} depends both on σ_{xy}^{VH} and R_L : $R_{NL} \propto \sigma_{xy}^{VH} R_L^3$. However, we find our calculation is broadly consistent with our experimental observation as we find nonlocal resistance when σ_{xy}^{VH} is nonzero and R_L is large. At first, we would like to clarify that in Supplementary Fig. 3 we had plotted σ_{xy} only for a high electric field value ($\Delta = 15$ meV). At such an electric field the CNP gap is large enough, but the moiré gaps are small. This is consistent with the fact that we do not observe nonlocal resistance in the moiré gaps for large electric fields. However, for the case of a low electric field as seen in Fig. R8, we do see nonzero σ_{xy} at moiré gaps also. This validates our observation of finite nonlocal resistance at the moiré gaps for low electric field values.

Fig. R8: σ_{xy} (on left axis) for a low electric field ($\Delta = 1$ meV) at $T = 5$ K (orange curve) and $T = 50$ K (blue curve). Corresponding density of states (DOS) showing moiré gaps (green curve corresponding to the right axis).

We would like to further mention that our observation is consistent with the fact that nonlocal resistance was observed in the moiré gaps of hBN aligned monolayer graphene [Gorbachev et al., *Science* **346**, 448 (2010)].

Action2: We have included the calculation of σ_{xy} for a lower electric field (as shown in Fig. R8) in the revised Supplementary Information (Supplementary Fig. 3). We appreciate the reviewer for pointing out that the reader might be interested in the R vs T curves for the moiré gap as well. We have included the following R vs T plot for the case of $n = -n_s$ in the revised Supplementary Information (Supplementary Fig. 13).

Fig. R9: Variation of Nonlocal (a) and local resistance (b) with temperature for the case of $n=-n_s$. The blue shaded background is the temperature regime used to show the scaling relation in Fig. 3e of the main manuscript.

Comment #3: --- Concerning the statement in page 7:

- > We note that the nonlocal signal at CNP originates
- > due to Berry curvature hotspots located at the edge of flat bands
- > (See Supplementary Sec. I for σ_{VH} at CNP in TDBG).

I feel emphasizing the "hot spots" may be confusing, because the nonlocal signal at CNP is related to the quantized in-gap valley Hall conductivity ($\sigma_{VH}=2e^2/h$) as plotted in Fig.S3, which is the integral of the Berry curvature all over the states below the gap.

Having a strong local Berry curvature at a particular k -point doesn't seem essential for observing non-local signals.

(it is just related to the derivative of σ_{VH} in energy)

Reply3: We agree with the reviewer that to observe non-local signals it is not essential for the Berry curvature to be localized at a particular k -point.

Action3: To remove possible confusion we have rewritten our statement without referring to 'hot spots'.

Indeed, the author states that R_{NL} is appreciable over larger range of charge density when the hotspot is more spread in k -space.

We thank the reviewer for correctly summarising one of the key observations in our experiment. We use the terminology of 'hot spot' in line with the existing literature. The experimentally measured R_{NL} in TDBG is appreciable mainly near the gaps, though over a *larger* range compared to other systems explored in the literature.

Minor points:

Comment #4: --- *Fig. 1d: provide Delta.*

Reply4 & Action4: We thank the reviewer for pointing this out. We have mentioned the value of Delta in Fig. 1d as well as in the caption in the revised manuscript.

Comment #5: --- *Fig. 3c and 3d: provide temperature.*

Reply5 & Action5: In Fig. 3c and 3d the gaps are extracted by fitting the Arrhenius equation to the R vs T curves in the temperature range within 15 K to 85 K. In the modified caption of Fig. 3c and Fig. 3d, we have mentioned this temperature range.

REVIEWERS' COMMENTS

Reviewer #1 (Remarks to the Author):

I think the authors carefully addressed all the comments in the revised manuscript and response letter. I support its publication in Nature Comm now.

Reviewer #2 (Remarks to the Author):

The authors have addressed all the important issues raised by the reviewer. The supplementary data provided and revisions in the main text are detailed and convincing. Especially, for Comment #4, the authors have successfully clarified the importance of the central findings of this work and it may have more far reaching inspirations for the explorations of topological Chern bands in the flat band systems, as well as valley current tailoring by external electric field. The overall novelty and quality of this manuscript has been substantially enhanced, and I recommend acceptance in Nature Communications.

Reviewer #3 (Remarks to the Author):

I found the authors' response and the revised version of the manuscript appropriately address the referees' questions. I recommend the paper for publication in Nature Communications.

Below, we respond to the reviewers' comments on the resubmitted manuscript:

(Reviewers' comments are in the *blue colored italic font* and our responses are in the black colored font.)

Response to Reviewer #1:

I think the authors carefully addressed all the comments in the revised manuscript and response letter. I support its publication in Nature Comm now.

We thank the reviewer for reviewing our manuscript and recommending the publication of the revised manuscript in Nature Communications. We are happy to have addressed the reviewer's important comments.

Response to Reviewer #2:

The authors have addressed all the important issues raised by the reviewer. The supplementary data provided and revisions in the main text are detailed and convincing. Especially, for Comment #4, the authors have successfully clarified the importance of the central findings of this work and it may have more far reaching inspirations for the explorations of topological Chern bands in the flat band systems, as well as valley current tailoring by external electric field. The overall novelty and quality of this manuscript has been substantially enhanced, and I recommend acceptance in Nature Communications.

We thank the reviewer for appreciating our study's novelty and highlighting its far-reaching aspects. We are additionally thankful for recommending the publication of the revised version in Nature Communications. We are happy that we have convincingly addressed the important questions raised by the reviewer.

Response to Reviewer #3:

I found the authors' response and the revised version of the manuscript appropriately address the referees' questions. I recommend the paper for publication in Nature Communications.

We thank the reviewer for reviewing our manuscript and for recommending to publish it in Nature Communications. We are happy to have addressed the reviewer's important questions.